# A new approach exploiting thermally activated delayed fluorescence molecules to optimize solar thermal energy storage

Fan-Yi Meng[1,2], I-Han Chen[1,2], Jiun-Yi Shen[1,2], Kai-Hsin Chang[1,2], Tai-Che Chou[1], Yi-An Chen[1], Yi-Ting Chen[1], Chi-Lin Chen[1] & Pi-Tai Chou [1✉]

We propose a new concept exploiting thermally activated delayed fluorescence (TADF) molecules as photosensitizers, storage units and signal transducers to harness solar thermal energy. Molecular composites based on the TADF core phenoxazine–triphenyltriazine (PXZ-TRZ) anchored with norbornadiene (NBD) were synthesized, yielding compounds PZDN and PZTN with two and four NBD units, respectively. Upon visible-light excitation, energy transfer to the triplet state of NBD occurred, followed by NBD → quadricyclane (QC) conversion, which can be monitored by changes in steady-state or time-resolved spectra. The small $S_1$-$T_1$ energy gap was found to be advantageous in optimizing the solar excitation wavelength. Upon tuning the molecule's triplet state energy lower than that of NBD (61 kcal/mol), as achieved by another composite PZQN, the efficiency of the NBD → QC conversion decreased drastically. Upon catalysis, the reverse QC → NBD reaction occurred at room temperature, converting the stored chemical energy back to heat with excellent reversibility.

[1] Department of Chemistry, National Taiwan University, R.O.C, Taipei 10617, Taiwan. [2] These authors contributed equally: Fan-Yi Meng, I-Han Chen, Jiun-Yi Shen, Kai-Hsin Chang. ✉email: chop@ntu.edu.tw

Worldwide energy consumption, which is predicted to double within the next 40 years, demands a shift toward widespread use of renewable energy[1]. Sunlight is a facilitative and inexhaustible energy source that can be harvested via photovoltaics[2,3] or thermal heating[4] for usage. One challenge in the development of solar energy technologies is the variation in both energy production and demand over time, which requires the use of load-leveling techniques[5]. While electricity can be stored using batteries for on-demand power supply, this technology faces challenges with respect to cost and large-scale implementation.

An alternative is the direct conversion from solar energy to stored chemical energy. This can be achieved, in principle, via the conversion of water to hydrogen[6] or the reduction of carbon dioxide to methanol[7], which, however, involves gaseous species. Energy storage can also be accomplished through photoisomerization[8]. A considerable number of molecular solar thermal (MOST) systems incorporating organic compounds, including anthracene[9,10], azobenzene[11–13], dihydroazulene[14,15], norbornadiene-quadricyclane[16–18], stilbene[19,20], and ruthenium fulvalene derivatives[21,22], which can undergo light-induced isomerization to metastable isomers for storage of solar energy, have been proposed.

Materials must fulfill a number of criteria before they can be suitable for practical MOST applications:[23] (a) The absorption spectrum has to be optimized with respect to the solar emission spectrum, known as the solar spectrum match, to achieve maximal efficiency. (b) The energy storage density should be high, which implies a compromise between storage energy and molecular size. (c) The barrier for the back conversion from the high-energy isomer to the low-energy isomer has to be sufficiently high to ensure long-term storage yet be accessible via a catalytic reaction to enable timely energy release. (d) The yield for either photoconversion or thermal conversion should be high, possibly 100%.

Among the abovementioned potential materials for MOST, the norbornadiene (NBD)-quadricyclane (QC) interconverting system, in which a large amount of energy can be stored in a small ring (QC) via photoconversion of NBD in the triplet state, has received considerable attention[24]. Unfortunately, the lowest lying absorption onset of the nonsubstituted NBD is no longer than 300 nm[25], and the energy of the transition forbidden triplet state of NBD is ~61 kcal/mol (~468 nm)[26]. To have practical application as a solar energy accumulator, chemical modification of NBD is needed to redshift the absorption wavelength to match that of sunlight reaching the earth (>400 nm)[27]. One strategy is to introduce donor and acceptor substitutions on the NBD skeleton and increase the π-conjugation length (see Fig. 1a)[28,29]. An alternative way is to provide a photosensitizer[30–34] whose absorption matches the solar radiation; in such a system, upon excitation, energy transfer can take place to populate the NBD triplet state, followed by NBD → QC conversion (see also Fig. 1a, b). The photosensitizer can chemically link NBD to allow intramolecular energy transfer to increase efficiency. To date, the development of photosensitizers to optimize MOST systems has been challenging, as the photosensitizer must have allowed electronic transitions similar to those of solar photons and a triplet state energy higher than that of NBD. In other words, this requirement implies that the energy between the $S_1$ and $T_1$ states is as close as possible for the photosensitizer to optimize energy harvesting. This viewpoint inspires the new concept of linking a photosensitizer to materials that exhibit thermally activated delayed fluorescence (TADF)[35–38] to achieve a suitable MOST system (see Fig. 1a). TADF molecules commonly possess spatially separated HOMOs and LUMOs. A small electron exchange integral leads to a small difference in energy, $\Delta E_{S-T}$, between the $S_1$ and $T_1$ states, leading to $T_1 \rightarrow S_1$ thermal repopulation and hence TADF. Since Adachi and coworkers[35] reported highly efficient OLEDs by harvesting the triplet state of TADF molecules, the quest to develop TADF molecules, understand their corresponding photophysics and identify applications in OLEDs has been one of the hottest research areas during the past decade.

Herein, taking advantage of the rather small, thermally achievable $\Delta E_{S-T}$ for TADF systems, we strategically designed a new series of MOST-relevant compounds using phenoxazine–triphenyltriazine (PXZ-TRZ) as a core. PXZ-TRZ has been reported to exhibit prominent TADF[39], with $T_1$ energies of ~63–65 kcal/mol estimated from the onset of phosphorescence (448–455 nm). The PXZ-TRZ core was found to effectively separate the HOMO and LUMO in a single molecule to obtain a near-zero-energy gap between $S_1$ and $T_1$[39]. Thus, the $T_1$ energy was semiquantitatively assessed through the observation of steady-state fluorescence. Upon anchoring the pendant NBD moiety, we observed slight changes in the associated PXZ-TRZ $S_1$ and $T_1$ energy levels, allowing fine-tuning of the state energy and hence optimization of the harvested solar energy. We also attempted to anchor as many NBDs as possible without considerable interference from the major chromophore to maximize the energy storage density. Accordingly, three PXZ-TRZ-NBD-type composites, PZDN, PZTN, and PZQN, were designed and synthesized (see Fig. 1c), among which PZDN and PZTN exhibit TADF that is strongly quenched by energy transfer and is thus dependent on the NBD → QC reaction time. In other words, both the TADF intensity and dynamics can be used to monitor the progress of the reaction. In sharp contrast, PZQN shows a nearly constant TADF rate where slow energy transfer occurs, retarding the NBD → QC reaction. These results show that subtle energy tuning dramatically alters the rate of the reaction, thus maximizing the energy transfer process. NBD → QC-dependent TADF spectroscopy and dynamics, together with reverse QC → NBD reaction thermodynamics, are elaborated for the first time in the following section.

## Results

**Synthesis.** Details of the syntheses and characterization are provided in the experimental section. In brief, the synthetic approach of the target molecules began with phenoxazine **1**. Formation of a C-N bond under Ullmann conditions with 4-bromoiodobenzene gave **2**[40] in moderate yield. Treating **2** with n-butyl lithium followed by reaction with cyanuric chloride yielded key intermediate **3**[41], which then reacted with carbon or oxygen nucleophiles to afford multiple norbornadiene-tethered species, PZDN, PZTN, and PZQN, with different linkers (see Fig. 1c). The norbornadiene derivative bicyclo[2.2.1]hepta-2,5-diene-2-methanol (NBD-OH) was prepared as reported in the literature[42]. This norbornadiene alcohol could either react with **3** directly or react with other linkers. Synthesis of PZTN was achieved through nucleophilic substitution of **3** with glycerol ether **4**. The latter was prepared with norbornadiene alcohol and epichlorohydrin[43]. It should be noted that the methyl group in reagent **5** is crucial in the synthesis of PZQN. If two hydrogens are present at the α position of malonate, the substitution intermediate with one unreacted chloride will be deprotonated on the tertiary carbon in addition to the carbonyl group. This is because the electron-withdrawing triazine moiety can increase the acidity of the α proton of the newly formed intermediate. As a result, the electron density on triazine will be dramatically increased by the carbanion to prevent subsequent nucleophilic substitution, and the reaction will be terminated with one chloride remaining.

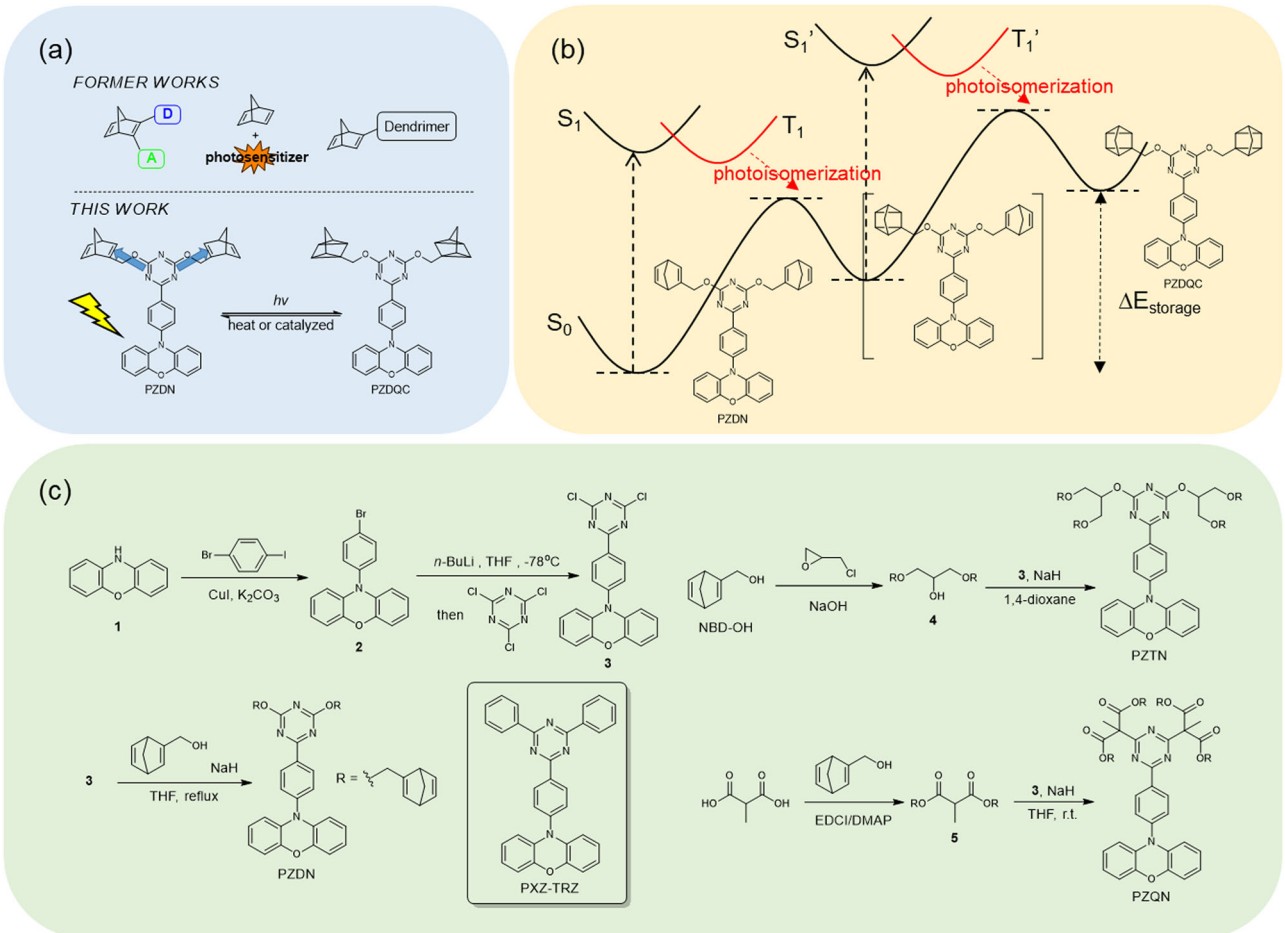

**Fig. 1 Recent developments in norbornadiene for energy storage and a novel concept in this work. a** Designed strategies to promote the efficiency of the MOST process in the NBD-QC system. **b** The proposed energy profile for the photochemical reaction of the PZDN-PZDQC process. **c** The synthetic route of PZDN, PZTN, and PZQN. Details of the reaction are provided in the supporting information.

**Photophysical properties**. Figure 2 shows the absorption and emission spectra of PZDN, PZTN, and PZQN in cyclohexane at room temperature. For the three title compounds, the lowest lying absorption peak at approximately 400–430 nm has charge transfer character, similar to that of PXZ-TRZ[39]. In addition, all three compounds were spectroscopically pure, supported by identical lowest lying bands in the excitation and absorption spectra (see Supplementary Fig. 15). To gain further insight into the electronic transition, computations were carried out with the Gaussian 16[44] package (see the Methods for details). The results shown in Fig. 3 (vide infra) clearly demonstrate that the lowest lying transition ($S_0 \rightarrow S_1$) for the title compounds possesses charge transfer character, where the HOMO and LUMO are mainly located on the phenoxazine unit and triazine moiety, respectively. Note that the NBD moieties do not contribute to either the HOMO or the LUMO. This finding explains the lack of obvious changes in the lowest lying absorption peak in the spectra of the title compounds and the core moiety PXZ-TRZ. Therefore, the NBD $\rightarrow$ QC conversion, if it occurs, is expected to be associated with the photoinduced energy transfer from the PXZ-TRZ analog to NBD (vide infra). Fig. 2 also displays the emission spectra of PZDN, PZTN, and PZQN. Despite the very similar absorption spectra, the emission spectra reveal a slight difference in peak wavelength; in particular, the emission maximum at ~530 nm in the spectrum of PZQN is redshifted by ~4.5 kcal/mol with respect to that (~490 nm) in the spectra of PZDN and PZTN. The results manifest various degrees of photoinduced

charge transfer, leading to different Stokes shifts in the emission and hence changes in the $S_1 \rightarrow S_0$ and $T_1 \rightarrow S_0$ energy gaps. The latter is crucial in terms of effective energy transfer to NBD (vide infra). Nevertheless, the three compounds all showed remarkable oxygen-dependent fluorescence intensity, which increased significantly in degassed solutions, such as cyclohexane, relative to aerated solutions, a typical TADF behavior also observed for the PXZ-TRZ core chromophore. That is, the energy of the lowest lying singlet state is in proximity to the triplet state in these molecules such that the $S_1 \rightarrow T_1$ intersystem crossing (ISC) and reverse intersystem crossing (RISC) are thermally reversible at room temperature.

**NBD $\rightarrow$ QC photoisomerization reaction**. The above-described absorption and emission spectra were acquired instantly with low radiation intensity. We thus assume no occurrence of photochemistry during the measurement. In degassed cyclohexane, upon irradiation with 405 nm light (~0.1 mW/cm²), as shown in Fig. 2, the emission intensity of PZDN gradually increased and reached a maximum of ~3.4 times the original intensity after approximately 1 h. A similar result was also observed in the photolysis of PZTN, except that the final intensity was approximately 1.8 times the original intensity. Additionally, the photophysics associated with the photochemistry of PZDN and PZTN can be more clearly observed via time-resolved emission spectroscopy, where after a designated irradiation period, the

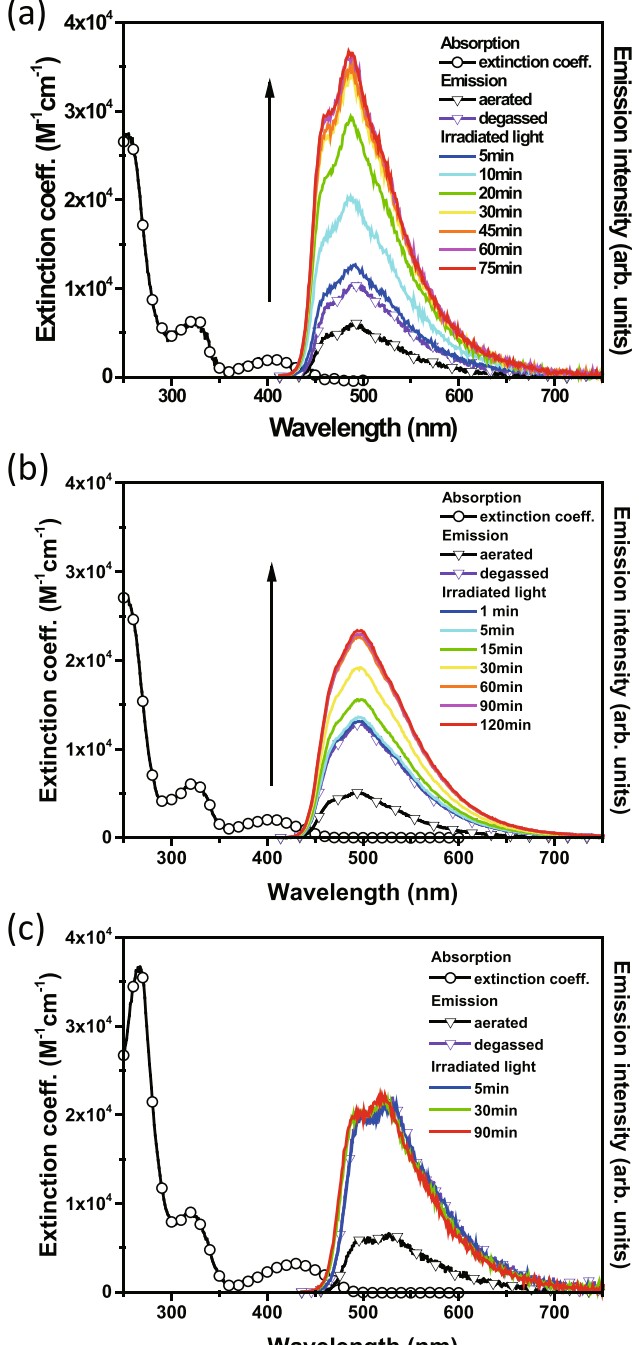

**Fig. 2 Irradiation time-dependent fluorescence spectra.** Absorption and emission spectra of PZDN (**a**), PZTN (**b**), and PZQN (**c**) are measured in cyclohexane ($\lambda_{ex} = 405$ nm). Note that the arrow represents the increment of emission intensity with photolysis time.

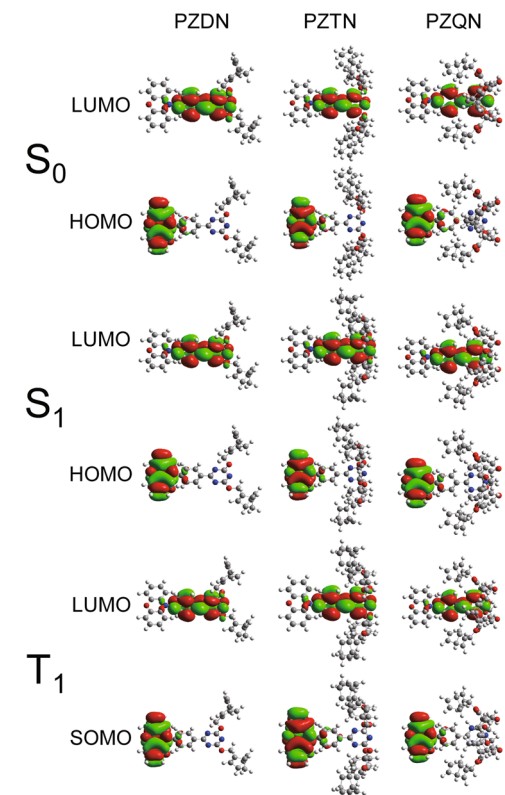

**Fig. 3 Optimized frontier orbital contours of PZDN, PZTN, and PZQN in the $S_0$, $S_1$, and $T_1$ states.** The $S_0$ state structures and excited state ($S_1$, $T_1$) structures were calculated by density functional theory (DFT) and time-dependent density functional theory (TD-DFT), respectively, with the PBE0 hybrid functional and the 6–31 G(d) basis set.

transient decay was acquired by time-correlated single-photon counting (TCSPC), which requires a very small photon flux to minimize the photoreaction. The results shown in Fig. 4 (vide infra) clearly reveal that the dynamics of the delayed fluorescence vary significantly as a function of irradiation time. In the degassed cyclohexane, immediately before irradiating PZDN, the population decay of the delayed fluorescence was fast and determined by fitting to be 0.7 μs (see green line in Fig. 4a), indicating quenching dynamics that can be well explained by intramolecular triplet (PXZ-TRZ core)-triplet (peripheral NBD) energy transfer. Upon

405 nm radiation, the decay of the delayed fluorescence gradually increased, accompanied by an increase in the steady-state TADF intensity (Fig. 2). At an irradiation time of, e.g., 90 min, the steady-state TADF intensity remained unchanged. The lifetime of the delayed fluorescence was determined by fitting to be as long as 10.8 μs (see the orange line in Fig. 4a and Supplementary Fig. 17). The increase in the area under the dynamic fitting curve was ~3.6 times. We thus reasonably conclude that following triplet-triplet energy transfer, NBD → QC isomerization occurred, forming the product denoted as PZDQC (see Fig. 4a). Note that DQC represents the formation of two moles of QC for the photolysis of each PZDN. As shown in Figs. 2b, 4b, similar changes in the steady-state emission intensity and relaxation dynamics as a function of irradiation time were also observed for PZTN, suggesting the same photophysics and photochemistry as PZDN, yielding the product named PZTQC because, in theory, each PZTN produces four moles of QC (vide infra). The increase in the area under the dynamic fitting curve was calculated to be ~2.2 times (PZTN → PZTQC), which also agrees with the result from steady-state measurements (~1.8 times). Furthermore, we carried out a photolysis experiment using a mixture of the methoxy-substituted analog PZDMe ($5 \times 10^{-5}$ M) and NBD-OH ($1.0 \times 10^{-4}$ M) in degassed cyclohexane (405 nm excitation, ~0.1 mW/cm$^2$). PZDMe and NBD-OH represent the separated sensitizer and acceptor parts, respectively. A stoichiometric ratio of 1:2 for PZDMe:NBD-OH was used to simulate each PZDN ($5 \times 10^{-5}$ M) having two NBD units. As a result, both the kinetic traces and steady-state spectra of PZDMe show negligible changes as a function of irradiation time (see Supplementary Fig. 16). The results suggest that intramolecular energy transfer is the major

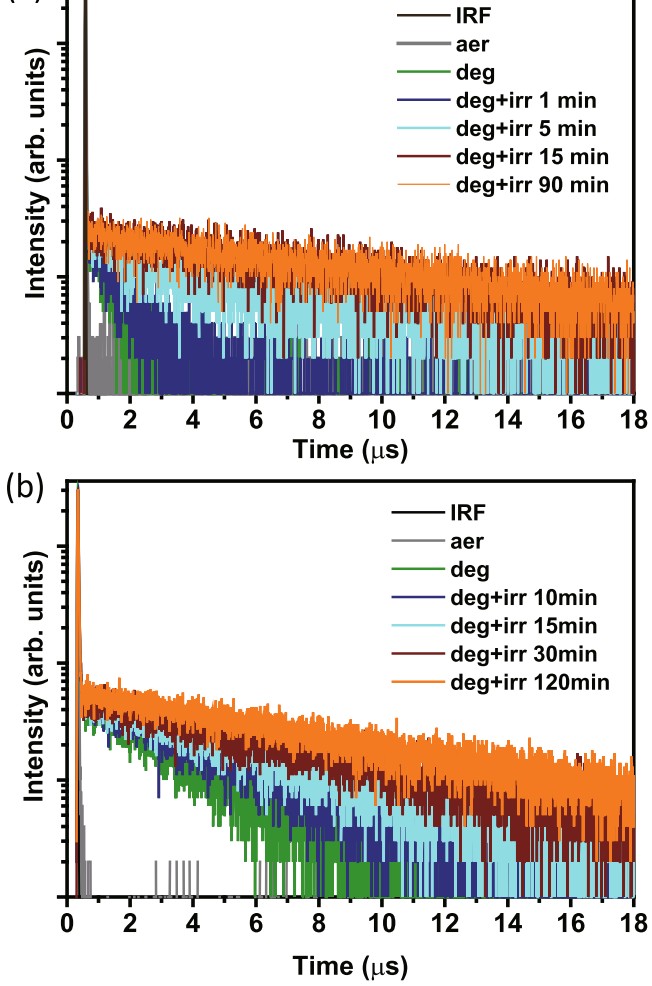

**Fig. 4 Irradiation time-dependent emission kinetic profiles.** The TADF decay dynamics of (**a**) PZDN and (**b**) PZTN as a function of irradiation (405 nm) time in room-temperature degassed cyclohexane. **"**aer", "deg" and "irr" refer to aerated conditions, degassed conditions and light irradiation conditions, respectively.

contribution to the energy transfer in PZDN, triggering the subsequent NBD → QC conversion.

We then carried out a dynamic simulation based on several valid assumptions[45,46] (see Supplementary Figs. 18 and 19 and the corresponding elaboration). These assumptions include a millisecond-scale decay of the lowest excited triplet state ($T_1$). Additionally, the prompt decay rate of the lowest excited singlet state ($S_1$) was approximated by the population decay of $S_1$. The deduced kinetic rates are tabulated in Supplementary Table 1. To quantify the photoconversion efficiency ($Q.Y._{eff}$), we further defined this term with Eq. (1), expressed as

$$Q.Y._{\cdot eff} = \frac{k_{ET}}{k_{ET} + k_{RISC} + k_P} \quad (1)$$

where $k_{ET}$, $k_{RISC}$, and $k_P$ refer to the rate constants of triplet-triplet energy transfer, reverse intersystem crossing ($T_1$ to $S_1$) and the radiative plus nonradiative decay of $T_1$, respectively. As a result, the simulated photoconversion efficiency was determined to be 59.4% for PZDN, 14.3% for PZTN, and 3.7% in PZQN. The results, on the one hand, can be rationalized by the distance between the PXZ-TRZ core and the energy acceptor NBD moiety, which is shorter in PZDN, resulting in more efficient Dexter-type energy transfer than that in PZTN. On the other hand, the energy

transfer for PZQN is thermally unfavorable (vide supra), and hence, the associated rate is drastically reduced. This trend correlates well with the change in fluorescence quantum yield before and after light illumination (see Table 1).

Chemically, firm evidence of the NBD → QC photoisomerization for PZDN and PZTN was provided by a proton NMR study. To study the photoisomerization of PZDN, an NMR tube containing PZDN in perdeuterated cyclohexane-$d_{12}$ was irradiated with a metal halide lamp (350–450 nm bandpass filter) and monitored with proton NMR spectroscopy to provide direct structural information (see Fig. 5a). As the irradiation time elapsed, as shown in Fig. 5b, a decrease in the characteristic signals of the NBD unit at 5.1 ppm (indicated by the blue arrow in Fig. 5b) was observed, accompanied by an increase in the characteristic signals of QC at 4.7 and 4.5 ppm (see red arrow). Also shown are the characteristic peaks of QC at 1.2–1.7 ppm (see Supplementary Fig. 4). The proton NMR study of the photoisomerization of PZTN was also carried out under similar conditions, and the results are shown in Fig. 6. As the irradiation time increased, the proton peaks located at 6.3 ppm and 6.6 ppm, which are assigned to vinyl protons on the norbornadiene moiety, decreased significantly. Simultaneously, the peaks from 3.6 to 3.8 ppm increased; these multiplet peaks resulted from the complex chemical surroundings of the α protons of the ether bonds. In brief, the [1]H NMR spectra confirmed that PZDN and PZTN were converted into PZDQC and PZTQC upon 350–400 nm excitation, and the photoisomerization was efficient due to the lack of any unexpected proton peaks, consistent with the results of steady-state measurements in the photoisomerization study (vide supra). As depicted in Fig. 1b, the PZDN → PZDQC photoconversion is a stepwise process for each NBD → QC conversion, the mechanism of which has been reported for an NBD → QC system with two NBD units attached[47]. Unfortunately, during photolysis, we did not observe [1]H NMR signals attributed to intermediates in which only one QC was formed (see Fig. 5), plausibly due to its short lifespan.

In sharp contrast, as shown in Fig. 2c, the emission intensity of PZQN was almost invariable after continuous irradiation for over 90 min in degassed cyclohexane. In addition, the changes in the TADF decay dynamics were negligible before, during, and after, e.g., 3 days of irradiation (see Supplementary Figs. 17, 20). The results are indicative of the retardation of triple-triplet energy transfer and hence a lack of NBD → QC isomerization. This phenomenon can be further verified by the simulation results (see Supplementary Fig. 19 and Table 1), in which the rate of triple-triplet energy transfer in PZQN is significantly slower than other competing rates by approximately two orders of magnitude.

The above discrepancy between PZDN (or PZTN) and PZQN is of fundamental importance. The emission onset of TADF for PZDN and PZTN is located at ~420 nm, i.e., ~68 kcal/mol for the $S_0$-$S_1$ energy gap. Moreover, the occurrence of TADF indicates that $\Delta E_{S-T}$ is < 5.0 kcal/mol. Therefore, the lower limit of the $T_1$ state for both PZDN and PZTN is estimated to be ~63 kcal/mol, which is higher than that of NBD (61 kcal/mol). The energy transfer from $T_1$ to the triplet of NBD is thermodynamically favorable. On the other hand, the emission onset of PZQN is approximately 460 nm in cyclohexane, corresponding to an $S_1$-$S_0$ energy gap of ~62.1 kcal/mol. Taking a maximum $\Delta E_{S-T}$ of ~5.0 kcal/mol allowed for the observation of TADF, the lower limit of the PZQN $T_1$ state was deduced to be 57.1 kcal/mol. From a chemistry point of view, the difference in the energy gap can be explained by the substitution position. Using the core moiety PXZ-TRZ as a reference, the NBD-substituted analogs PZDN and PZTN, with electron-donating ether linkers, increase the energy of the LUMO of the 1,3,5-triazine moiety (see Fig. 3)[48]. The net result is to increase the $S_0$-$S_1$ energy gap and likewise the $S_0$-$T_1$

**Table 1 MOST properties in the solution phase.**

| Name | Abs. peak[a] (nm) | Abs. onset[a] (nm) | Fluo. peak[a] (nm) | Phos. onset[b] (nm) | $\Delta E_{ST}$[c] (kcal/mol) | Q.Y.$_{eff}$[d] (%) | Q.Y.$_{fluo}$[e] (%) | M.W. (g mol$^{-1}$) | $\Delta H_{storage}$ (kJ mol$^{-1}$) | $\Delta H_{storage}$ (J g$^{-1}$) | $\tau_{1/2}$[f] (days) |
|---|---|---|---|---|---|---|---|---|---|---|---|
| PZDN | 405 nm | 465 nm | 495 nm | 454 nm | 0.85 | 59.4 | 11.0[g] 20.5[h] 66.9[i] | 578.23 | 162 | 280 | 76 |
| PZTN | 405 nm | 465 nm | 495 nm | 454 nm | 0.76 | 14.3 | 14.7[g] 37.8[h] 71.4[i] | 934.43 | 7.97 | 8.53 | –[j] |
| PZQN | 430 nm | 495 nm | 525 nm | 481 nm | 1.97 | –[k] | 20.0[g] 66.1[h] 66.1[i] | 988.40 | –[k] | –[k] | –[k] |

[a]Absorption (Abs.) and fluorescence (Fluo.) spectra measured at 298 K in cyclohexane.
[b]Phosphorescence (Phos.) spectra measured at 77 K in cyclohexane.
[c]$\Delta E_{ST}$ is calculated by the energy difference between the fluorescence and phosphorescence peaks at 77 K.
[d]The quantum yield for photoconversion is calculated by plugging the simulated rate constants (see Supplementary Table 1) into Eq. (1).
[e]Fluorescence quantum yield measured at 298 K in cyclohexane.
[f]Half-life of QC was obtained by extrapolating the Arrhenius plot (Supplementary Fig. 25) to 298 K.
[g]Measured under aerated conditions before irradiation.
[h]Measured under degassed conditions before irradiation.
[i]Measured under degassed conditions after irradiation.
[j]PZTQC → PZTN is too slow to allow practical measurements at 95 °C.
[k]Energy transfer can be ignored.

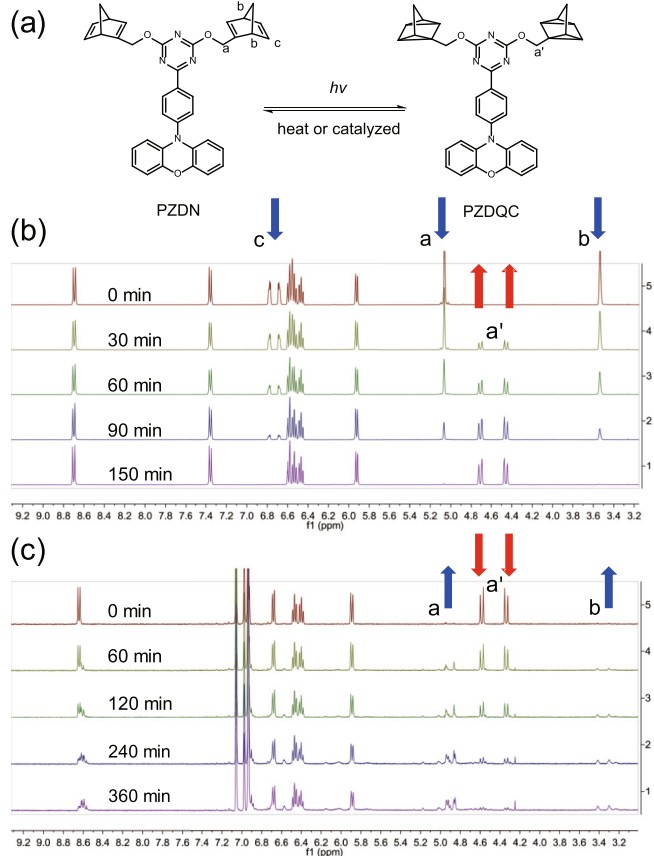

(a)

(b)

(c)

**Fig. 5 PZDN → PZDQC photoisomerization and PZDQC → PZDN reverse thermal (or catalytic) isomerization. a** Illustration of the reactions. **b** Time-dependent ¹H NMR spectrum of PZDN → PZDQC photoisomerization in cyclohexane-d₁₂ upon exposure to a metal halide lamp with a bandpass filter between 350 and 450 nm. **c** ¹H NMR spectrum of PZDQC → PZDN thermal isomerization by heating at 95 °C as a function of time (in toluene-d₈). Note that (**b**) and (**c**) are in different solvents (see text for the explanation), cyclohexane-d₁₂ and toluene-d₈, respectively, resulting in slightly different ¹H NMR peak positions. The blue and red arrows indicate the characteristic peaks for NBD and QC, respectively.

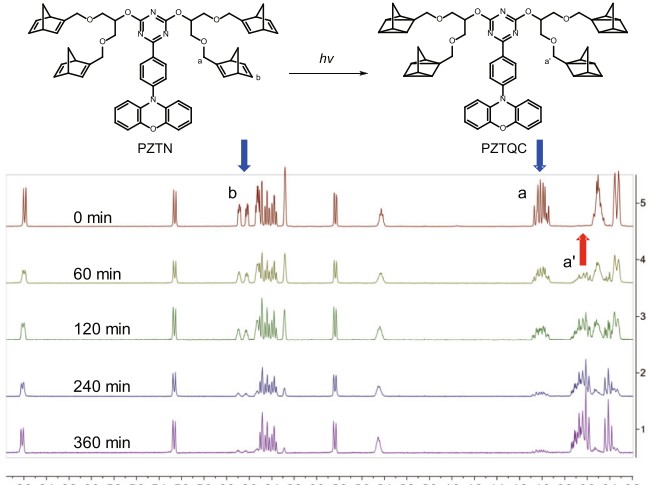

**Fig. 6 PZTN → PZTQC photoisomerization.** ¹H NMR spectrum of PZTN photoisomerization to PZTQC by exposure to a metal halide lamp (a bandpass filter between 350 and 450 nm) as a function of time (in cyclohexane-d₁₂).

energy gap for PZDN and PZTN (cf. PXZ-TRZ). Conversely, PZQN possesses an alkyl bridge linked by an ester substituent, which is considered a weak electron-withdrawing group. The net result is to decrease the LUMO energy and hence to decrease the $S_0$-$S_1$ and $S_0$-$T_1$ energy gaps. This chemical relationship is consistent with the experimentally observed prominent energy transfer to NBD for PZDN and PZTN, whereas such energy transfer is inefficient for PZQN because its $T_1$ energy is lower than that of NBD.

To gain further insight, we then attempted to estimate the triplet state energy by measuring the corresponding phosphorescence in a 77 K cyclohexane matrix. The phosphorescence spectra showed a trend of peak wavelength in the order of PZQN (590 nm) > PZDN (556 nm) ~ PZTN (554 nm) (see Supplementary Fig. 21 and Table 1). The onset of phosphorescence and hence the triplet state energy at 77 K PZDN and PZTN was approximately 63 kcal/mol (see Supplementary Fig. 21), which is

higher than that of the NBD triplet state at room temperature (61 kcal/mol). On the other hand, the onset of phosphorescence of ~59 kcal/mol (see Supplementary Fig. 21) for PZQN makes energy transfer to NBD thermally unfavorable, which drastically decelerates NBD → QC photoisomerization. One can thus precisely harness the triplet state energy and bring the MOST compounds close to NBD's triplet state. This strategy of using a TADF photosensitizer promotes the utilization of the solar spectrum, maximizing the harvest of solar energy for NBD → QC photoisomerization.

**QC → NBD reverse isomerization.** After the success of forward chemical storage via energy transfer and photoisomerization, we then investigated the release of chemical energy via a reverse reaction, i.e., QC → NBD isomerization. When PZDN was completely converted, as indicated by the disappearance of the PZDN proton peak (5.1 ppm) and saturation of the PZDQC proton peaks (4.7 and 4.5 ppm, vide supra), the product PZDQC was stable in room-temperature solution (cyclohexane $d_{12}$, NMR tube) since there were no changes in the corresponding NMR spectra within 24 h. To probe the thermally activated reverse QC → NBD isomerization by proton NMR, we then discarded the low-boiling-point (80.75 °C) cyclohexane-$d_{12}$ and instead used toluene $d_8$ (boiling point 110.6 °C), where the NMR tube was heated to 368 K (95 °C), to observe measurable changes in the NMR spectra. As the heating continued, as shown in Fig. 5c, the characteristic signals of the oxymethylene quadricyclane unit at ~4.7 and ~4.5 ppm gradually decreased, accompanied by an increase in the representative signal of the corresponding NBD unit at 5.1 ppm. These results demonstrate the feasibility of reversing the PZDQC → PZDN conversion via thermally activated isomerization. Temperature-dependent isomerization of PZDQC was then performed to determine the kinetic parameters of the reverse reaction. Supplementary Fig. 24 shows the logarithm of the ratio for time-dependent QC concentration versus $QC_0$ (at $t = 0$) of PZDQC by monitoring the integrated intensity of the 4.5 ppm peak during the reaction at 348 K, 358 K, and 368 K in toluene-$d_8$. As a result, the reverse rate, $k_{rev}$, of PZDQC → PZDN was obtained at each temperature. Supplementary Fig. 25 shows a plot of the logarithm of $k_{rev}$ versus as a function of the reciprocal of temperature according to the Arrhenius Eq. (2).

$$\ln(k_{rev}) = \ln(A) - \frac{E_a}{RT} \qquad (2)$$

where $E_a$ is the reaction activation energy and $A$ is the frequency factor. As a result, for PZDQC, the activation energy $E_a$ for thermal reverse isomerization was deduced to be as large as 23.59 ± 0.5 kcal/mol in toluene-$d_8$. To determine the transition state thermodynamics, a plot for the logarithm of $k_{rev}/T$ versus the reciprocal of temperature according to Eq. (3) is shown in Supplementary Fig. 26.

$$\ln\left(\frac{k_{rev}}{T}\right) = -\frac{\Delta H^\dagger}{R}\frac{1}{T} + \ln\left(\frac{k_B}{h}\right) + \frac{\Delta S^\dagger}{R} \qquad (3)$$

where $\Delta H^\dagger$ and $\Delta S^\dagger$ are the enthalpy and entropy, respectively, of the transition state. $k_B$ is Boltzmann's constant, and $h$ is Planck's constant. As a result, for PZDQC → PZDN, $\Delta H^\dagger$ and $\Delta S^\dagger$ were calculated to be 97.14 ± 4.9 kJ/mol and −53 ± 2.1 J/mol, respectively.

Comparatively, we also carried out the thermally activated PZTQC → PZTN reverse reaction. However, the reaction rate was found to be much slower than that of PZDQC → PZDN at an elevated temperature of 95 °C. The QC → NBD reverse reaction has been reported to be substituent dependent[49]. Thus, the

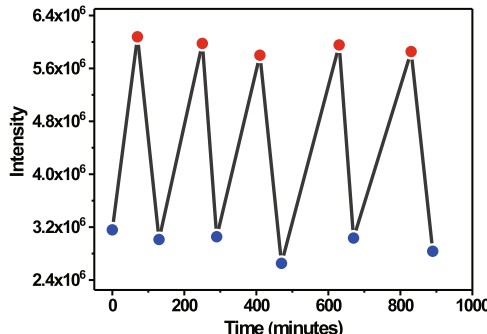

**Fig. 7 The durability of degassed PZDN ( ● ) ⇆ PZDQC ( ● ).** The experiment proceeded in cyclohexane, and the emission intensity at ~495 nm was monitored.

reverse PZTQC → PZTN reaction may require a higher activation energy. This, together with the more flexible sidechain of PZTQC, may lead to certain intramolecular decomposition pathways (see Supplementary Fig. 30 and vide infra). For practical applications, we thus alternatively sought a suitable catalyst for the reverse reaction. Two factors must be considered in this approach. First, the reduction of the energy barrier can prevent side reactions. Second, the release energy instantaneously fulfills the purpose of using the stored energy. We then carried out the reverse reaction for both PZDQC and PZTQC using cobalt tetraphenylporphyrin (CoTPP) as the catalyst[50,51]. In this experiment, a catalytic amount of CoTPP was added to a PZDQC (or PZTQC) solution (in toluene-$d_8$) under vigorous stirring at room temperature. The results shown in Supplementary Figs. 27 and 28 clearly indicate the quantitative yield of PZDN and PZTN by analyzing the growth of the corresponding NBD proton NMR peak.

Moreover, a durability test was also performed on PZDN to examine the energy storage-releasing process, i.e., the photoisomerization (PZDN → PZDQC) and reverse thermal isomerization (PZDQC → PZDN) cycles. The difference in absorption between PZDN and PZDQC was negligible. However, the TADF emission intensity of PZDN shows significant quenching due to the energy transfer to NBD. In addition, during the reaction cycle, no unwanted emission was observed. Therefore, for convenience, the emission peak intensity was utilized to probe the durability of this reversible process. Fig. 7 shows the plot of the emission peak intensity for PZDN and PZDQC during the forward-backward reaction. Fig. 7 clearly demonstrates the excellent fatigue resistance after five reaction cycles. These results show that PZDQC undergoes a significant QC → NBD reverse reaction at a relatively low temperature of 50–70 °C, which is the main reaction channel before generating any side products. Therefore, the results show that PZDN ↔ PZDQC interconversion is highly reversible. In comparison, PZTQC → PZTN conversion is much slower, so a certain branching reaction is competitive, yielding side products with relatively low reversibility.

Although the high exothermic energy involved in the QC → NBD reaction has been well established, for confirmation, we still performed the relevant experiment to determine the reaction thermodynamics. This was achieved by applying differential scanning calorimetry (DSC) measurements (see the SI for detailed experimental information). The results showed that PZDQC gave rise to one exothermic peak upon isomerization to PZDN, which was calculated to be −161.8 kJ/mol or −280.0 J/g (Supplementary Fig. 29). The ΔH value for PZDQC corresponded to approximately 2 moles reacting in the QC → NBD reaction (−89.00 kJ/mol or −153.92 J/g)[29,52], consistent with the reaction stoichiometry. The DSC result of PZTQC showed only a weak exothermic

peak at higher temperatures (>140 °C, see Supplementary Fig. 30), with a severe decrease in the integrated value. This result implies that some thermal decomposition process occurs together with the QC → NBD reaction. This explains the difficulty in the reverse PZTQC → PZTN reaction carried out at 95 °C (vide supra) (synthesis of enough amount of PZTQC for measurement with oxygen bomb calorimeter is a challenge work).

Finally, we realized that the urgently needed MOST materials should be prepared in a condensed state. Therefore, we conducted independent photoconversion experiments on PZDN in the pure solid state (solid film) and PZDN embedded in various polymer films, including polyethylene (PE), polystyrene (PS), polyvinyl chloride (PVC) and poly(methyl methacrylate) (PMMA). Unfortunately, all tested films underwent severe intermolecular aggregation even at low doping concentrations of 0.1 wt% or less, resulting in inferior conversion efficiency and durability. Relevant works are shown in Supplementary Figs. 31–35 with detailed descriptions.

## Discussion

In summary, we strategically designed and synthesized a series of new chemical storage compounds, PZDN, PZTN, and PZQN, bearing the TADF PXZ-TRZ core as a triplet state energy transmitter. The triplet energy can be straightforwardly estimated by means of the onset of room-temperature fluorescence due to the rather small $\Delta E_{S-T}$. As a result, PZDN and PZTN maximize the harvest of solar energy to trigger the forward NBD → QC reaction by PZDN (or PZTN) → NBD triplet-triplet energy transfer. The precision of energy tuning is further supported by PZQN, where the slight redshift of the TADF emission (cf. PZDN and PZTN) slows the triplet-triplet energy transfer and thus retards the NBD → QC conversion. The storage capacity for PZDN has been determined to be equivalent to two NBD units with excellent durability in the PZDN ⇆ PZDQC cycle. We thus report a new and practical approach exploiting thermally activated delayed fluorescence molecules, which act as photosensitizers, storage units, and signal transducers, to optimize solar thermal energy storage. This new strategy could broaden the choice of both TADF chromophores and the number of chemically modified norbornadiene moieties to maximize solar thermal energy storage. Further practical performance in solid devices may be boosted through structural modification, such as the introduction of sterically bulky substituents to prevent severe aggregation and elongation of the NBD triplet state energy toward red, along with spectrally matched TADF molecules, so that sunlight harvesting can be maximized.

## Methods

**Materials**. THF was dried over sodium/benzophenone and distilled water prior to use. Dichloromethane and 1,4-dioxane were dried over $CaH_2$ and distilled water prior to use. n-Hexane was purchased from Merck and used without purification, and 2.5 M n-BuLi in hexane was purchased from Acros and stored in the Acroseal package. NaH was purchased from Acros and stored in an $N_2$ atmosphere. All other reagents were purchased from Acros, TCI, or AK Scientific and used without further purification.

**Instrumentation**. NMR spectra were recorded on a Bruker Avance 400 or Varian Unity 400 spectrometer (1H, 13 C, 400, and 100 MHz, respectively) in appropriate deuterated solvents. High-resolution mass spectra were obtained with a Waters LCT Premier XE with an ESI source or a JEOL JMS-700 with an FAB source. DSC analysis was performed with Netzsch 204 F1. Steady-state absorption and emission spectra were recorded by a Hitachi (U-3310) spectrophotometer and an Edinburgh (FS920 and FS980) fluorimeter, respectively. Nanosecond time-resolved studies were performed by means of an Edinburgh FL 900 TCSPC system with a pulsed hydrogen-filled lamp as the excitation light source and an Edinburgh FL980 TCSPC system coupled to a silicon diode laser pulse. Phosphorescence spectroscopic measurements were carried out by means of a Princeton Instruments PIMAX system (CCD).

**Photophysical Measurements and simulation**. Steady-state absorption and emission spectra were recorded by means of a Hitachi (U-3310) spectrophotometer and an Edinburgh (FS920 and FS980) fluorimeter, respectively. The fluorescence quantum yields for PZDN, PZTN, and PZQN in cyclohexane were measured at 298 K and compared to those of Coumarin 480 in methanol (Q.Y. = 0.87). The irradiation source for the photochemistry analysis was chosen to be a 450 W Xe lamp (or 405 nm 3 W LED lamp) coupled with a monochromator with a slit open to allow an ~10 nm bandwidth at the selected wavelength (~0.1 mW). Nanosecond time-resolved population decays were fitted with the sum of exponential functions using the nonlinear least-squares procedure in combination with reconvolution fitting with the instrumental response function (IRF). The phosphorescence at 77 K was measured by means of a Princeton Instruments PIMAX system in conjunction with an intensified charge-coupled detector (ICCD), for which the time gating was delayed to the phosphorescence domain. The dynamic simulation was carried out by means of *Wolfram Mathematica 10.1*[53].

**Computational approach**. All computations were carried out by means of the Gaussian 16 package. The structure of the ground state ($S_0$) was optimized by density functional theory (DFT), and the excited state ($S_1$ and $T_1$) was calculated by time-dependent density functional theory (TD-DFT) with the PBE0[54,55] hybrid functional. The 6–31 G(d) basis set[56,57] was employed for all atoms. Calculations of both the ground and excited states in cyclohexane incorporated a polarizable continuum model (PCM)[58].

**Sample preparation for photoisomerization measurements**. First, the compound was dissolved in cyclohexane-$d_{12}$ and then degassed and transferred into a valved NMR tube to prevent influences from $O_2$. After that, the NMR tube was irradiated for different times (minutes) by a metal halide lamp.

**Reverse reaction for PZDQC and PZTQC with CoTPP as the catalyst**. PZDQC or PZTQC was dissolved in toluene (5 mL), CoTPP (1 mg) was added, and the mixture was stirred at room temperature for 4 h. After that, the solvent was removed in vacuo, and the residue was chromatographed on silica gel to afford the corresponding PZDN or PZTN in quantitative yield.

**DSC measurement**. PZDN was dissolved in cyclohexane, which was degassed to remove $O_2$. After radiation, the PZDN was wholly converted to PZDQC, and then the solvent was removed. The DSC experiment was performed from 303.15 K to 423.15 K with a 2 K/min heating rate in an $N_2$ atmosphere.

**The durability of degassed PZDN ⇆ PZDQC**. A solution of PZDN in cyclohexane was degassed by three consecutive freeze-pump-thaw cycles. Then, the degassed cuvette was irradiated with a 450 W Xenon lamp until the emission intensity of the PZDN no longer increased. Then, the irradiated sample was heated on a hot plate until the emission intensity of the PZDN was reversed (heated at 87 °C). The durability was evaluated from the differences in the emission intensity values between the NBD moieties and the QC moieties on the first and *n*th cycles of the reactions.

## Data availability

Most data generated or analyzed during this study are included in this published article or the Supplementary Information. All data are available from the authors upon reasonable request.

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

## Acknowledgements

P.-T.C. thanks the Ministry of Science and Technology, Taiwan for financial support. We thank the Proteomics MS Core Facility at Department of Chemistry, National Taiwan University. Mass spectrometry analyses were performed by Mass Spectrometry facility of the Institute of Chemistry, Academia Sinica, Taiwan. Thanks for the DSC data from Thermal Analysis System of Instrumentation Center, National Taiwan University, and they are the BEST Thermal Analysis System in Taiwan.

## Author contributions

P.-T.C. designed and leaded the investigations. F.-Y.M., I.-H.C., and J.-Y.S. performed the synthesis of titled molecules. F.-Y.M., I.-H.C., and J.-Y.S. performed the durability

experiments. K.-H.C., Y.-A.C., Y.-T.C., and T.-C.C. performed steady-state and time-resolved photophysical experiments. F.-Y.M. and C.-L.C. performed the DFT calculations. K.-H.C. and T.-C.C. performed low-temperature phosphorescence experiment. T.-C.C. performed numerical analysis on the rate constants.

## Competing interests
The authors declare no competing interests.
