## [Peer Review File · Nature Communications]

REVIEWER COMMENTS

Reviewer #1 (Remarks to the Author):

In this manuscript, Chou and co-workers describe the development of a unique energy storage system by making the use of photo/thermal chemical reaction of norbornadiene (NBD)-quadricyclane (QC) sensitized by thermally activated delayed fluorescence (TADF) unit. The designed compounds PZDN, PZTN, and PZQN are readily synthesized and characterized. It is noteworthy that the PZDN and PZTN show efficient energy-transfer from D–A unit, which serves as TADF luminophore, to the NBD, allowing for photochemical reaction of NBD moiety to QC. This chemical conversion was surely monitored with NMR spectroscopy. The authors demonstrated the time-resolved spectroscopic analysis of the synthesized compounds to reveal the irradiation time dependency of their TADF emission. Furthermore, the thermal analysis of the compounds quantitatively confirmed the occurrence of double photochemical reaction in PZDN. Since the storage of solar energy in a chemical form is highly challenging task, the research work presented herein provides a new avenue toward efficient storage method of converting photon energy into chemical energy. Overall, the manuscript is concisely presented, and the results and discussion are clear and sounds. Therefore, I would love to recommend the publication of this manuscript in Nature Communications, given the authors address the following minor points.

1. As for the purity of the final compounds (PZDN, PZTN, and PZQN), the authors checked only NMR and HR-MS, which are not sufficient for photophysical analysis. The authors might want to check the purity again with elemental analysis or excitation spectra.
2. The authors stated that the D–A unit in PZDN and PZTN serve as sensitizer to efficiently convert NBD unit into QC unit. In comparison with PZQN, it is clear that the presence of such a TADF-active unit inside molecule is important for energy transfer. But, the authors do not compare with intermolecular process. It would be nicer to compare the photochemical conversion efficiency of PZDN/PZTN with two-component system (NBD and sensitizer such as PXZ-TRZ).
3. Showing the energy profile of photochemical reaction of PZDN-PZDQC in manuscript would be very helpful for readers.

Reviewer #2 (Remarks to the Author):

In this manuscript, Chou et al. report synthesis of three norbornadiene (NBD) derivatives by exploiting thermally activated delayed fluorescence molecules (TADF). The TADF may act as a

photosensitizer, storage unit and signal transducer to optimize solar thermal energy storage in the novel molecules. The NBD and its derivatives have been widely studied for MOST. Although the molecular design is interesting, these results cannot advance the progress of solar thermal energy storage materials. I do not suggest acceptance of the manuscript at the present format.

- 1) In Figure 3b,c, I cannot find the emission in the degassed state (maybe purple curve?) .
- 2) The urgent topic of MOST materials should be performed in condensed state, not in the solution state.
- 3) The small S1-T1 energy gap offers advantage in optimizing solar excitation wavelength, however it only covers a narrow arrange in the solar spectrum.
- 4) Only DSC was provided in Supplementary Figure 24, which is not enough for solar thermal energy storage materials. The MOST device should be studied based on charging and recharging processes.

Reviewer #3 (Remarks to the Author):

The Manuscript " A New Approach Exploiting Thermally Activated Delayed Fluorescence Molecules to Optimize Solar Thermal Energy Storage" by Pi-Tai Chou and co-workers considers a molecular photoswitch system capable of converting photon energy into stored chemical energy – so-called molecular solar thermal systems. The novelty of the molecular system is related to the chromophore design, that incorporates a TADF component into the molecular system. Further, the energy storing capability of the system is augmented by the attachment of several NBD/QC units to the central chromophore unit.

The basic idea of the manuscript is to drive the photoisomerisation through the triplet manifold via energy transfer from a TADF unit.

The introduction is well written with extensive references to the literature.

The description of the synthesis is well written, and I appreciated the pedagogic description of the challenges in the synthesis design together with actual synthesis results.

The experimental work is complemented with DFT based modelling, trying to understand the positioning of the orbitals.

The system is further studied using a combination of time resolved spectroscopic techniques.

All together, I think that this is an exciting manuscript that deserves publication in nat. commun. Pending some revisions as noted:

Major Comments

1) The fluorescence is competing with the photoconversion, what is the quantum yield of the two processes?

Minor comments:

1) The spelling and language should be checked, e.g. in the abstract: "triple state of" should be "triplet state"

2) "fMOST" should read "MOST"

3) "implies as closer energy"

4) To help the presentation, I suggest to make a table that summarizes the properties of the 3 molecular systems, e.g. λ_{max} , λ_{onset} , QY(photoconversion), QY(emission), T_{0.5}, Estorage(J/mol), Estorage (J/kg). such a table would make comparison with other systems much more straight forward, see e.g. review by G Han: <https://doi.org/10.1039/D1TC01472B>

5) The figures could in general be more refined, the NMR spectra are formatted in different ways (fig. 6 vs fig. 7, fig. 5 looks terrible with the legend text, etc. please check the journal guidelines, this is a quality journal, there should be quality figures.

6) Why only one compound in fig 8? I suggest to put this in the SI and measure for all compounds. The suggested table can give the needed summary. Check meaningful digits!!

7) DSC measurements of the energy storage should ideally be provided for all 3 compounds, and the measurements repeated atleast 2 times for each, due to the uncertainty of DSC measurements.

RESPONSE TO REVIEWERS

Reviewer 1: In this manuscript, Chou and co-workers describe the development of a unique energy storage system by making the use of photo/thermal chemical reaction of norbarnadiene (NBD)-quadricyclane (QC) sensitized by thermally activated delayed fluorescence (TADF) unit. The designed compounds PZDN, PZTN, and PZQN are readily synthesized and characterized. It is noteworthy that the PZDN and PZTN show efficient energy-transfer from D–A unit, which serves as TADF luminophore, to the NBD, allowing for photochemical reaction of NBD moiety to QC. This chemical conversion was surely monitored with NMR spectroscopy. The authors demonstrated the time-resolved spectroscopic analysis of the synthesized compounds to reveal the irradiation time dependency of their TADF emission. Furthermore, the thermal analysis of the compounds quantitatively confirmed the occurrence of double photochemical reaction in PZDN. Since the storage of solar energy in a chemical form is highly challenging task, the research work presented herein provides a new avenue toward efficient storage method of converting photon energy into chemical energy. Overall, the manuscript is concisely presented, and the results and discussion are clear and sounds. Therefore, I would love to recommend the publication of this manuscript in Nature Communications, given the authors address the following minor points.

Reply: We are very grateful to the reviewer for his/her positive comments on this manuscript.

Minor issues:

1. As for the purity of the final compounds (PZDN, PZTN, and PZQN), the authors checked only NMR and HR-MS, which are not sufficient for photophysical analysis. The authors might want to check the purity again with elemental analysis or excitation spectra.

Reply: We thank the reviewer for his/her valuable comments. As for the photophysical properties, we have added the excitation spectrum of PZDN, PZTN, and PZQN (in cyclohexane) in supporting information (see **Supplementary Fig. 15**), which is also shown below for the reviewer's convenience.

Supplementary Figure 15. The excitation spectra of (a) **PZDN**, (b) **PZDN** and (c) **PZTN** in cyclohexane at room temperature. Note: “em 500” refers to the monitored emission wavelength. Mismatches between absorption and excitation spectra in higher lying region of 250 ~ 350 nm originate from the combination of fast non-radiative decays from highly excited state and inner filter effect.

In comparison, the excitation spectra and absorption in the lowest energy bands are identical, indicating that **PZDN**, **PZTN** and **PZQN** are all spectroscopically pure. Note that certain mismatches between absorption and excitation spectra in higher lying region of 250 ~ 350 nm originate from the combination of fast non-radiative decays from the highly excited state and inner filter effect. The following statement has been added in the revised text (see page 2, right column, line 4):

“In addition, all three compounds were examined to be spectroscopically pure, supported by the identical excitation and absorption spectra in the lowest lying band (see Supplementary Fig. 15).”

2. *The authors stated that the D–A unit in PZDN and PZTN serve as sensitizer to efficiently convert NBD unit into QC unit. In comparison with PZQN, it is clear that the presence of such a TADF-active unit inside molecule is important for energy transfer. But, the authors do not compare with intermolecular process. It would be nicer to compare the photochemical conversion efficiency of PZDN/PZTN with two-component system (NDB and sensitizer such as PXZ-TRZ).*

Reply: We thank the reviewer for pointing out a possible intermolecular energy transfer process. Accordingly, we have performed an experiment where we mixed **PZDMe** (5×10^{-5} M) and **NBD-OH** (1×10^{-4} M, see structure below) in cyclohexane. **PZDMe** and **NBD-OH** represent the separated sensitizer and acceptor parts, respectively. The prepared stoichiometric ratio 1 : 2 for **PZDMe** : **NBD-OH** in concentration is to simulate **PZDN**. As a result, both kinetic traces and steady state spectra as a function of irradiation time (405 nm light, ~ 0.1 mW/cm²) in degassed cyclohexane are added in the supporting information, **Supplementary Fig. 16**. For the reviewer’s convenience, this newly added figure is also shown below:

Supplementary Figure 16. Intermolecular sensitization of PZDMe (5×10^{-5} M) and NBD-OH (1×10^{-4} M) mixture in cyclohexane at room temperature. (a) The chemical structure of PZDMe and NBD-OH. (b) Steady state measurements ($\lambda_{\text{ex}} = 400$ nm) and (c) kinetic trace (excitation wavelength: 378 nm) under degassed condition as a function of irradiation time ($\lambda_{\text{ex}} = 405$ nm, ~ 0.1 mW/cm²).

As shown in **Supplementary Fig. 16**, clearly, after 10 minutes of irradiation, the steady state emission revealed negligible increase of intensity. This is also supported by the time-resolved measurement where the emission decay kinetics are about the same before and after 405 nm irradiation. In comparison to **PZDN** where the significant increase was observed in both steady state emission intensity and lifetime of the delayed fluorescence (see **Fig. 2a** in text), we thus conclude that the energy transfer process of **PZDN**, triggering the subsequent chemical conversion, is mainly an intramolecular process in this study. As a result, the following statement has been added in the revised manuscript (see page 4, line 6):

“Furthermore, we carried out the photolysis experiment using a mixture of methoxy-substituted analogue **PZDMe** (5×10^{-5} M) and **NBD-OH** (1.0×10^{-4} M) in the degassed cyclohexane (405 nm excitation, ~ 0.1 mW/cm²). **PZDMe** and **NBD-OH** represent the separated sensitizer and acceptor parts, respectively. The prepared stoichiometric ratio 1 : 2 for **PZDMe** : **NBD-OH** in concentration is to simulate each **PZDN** (5×10^{-5} M) having two NBD units. As a result, both kinetic traces and steady

state spectra of **PZDMe** show negligible changes as a function of irradiation time (see Supplementary Fig. 16). The results support that intramolecular energy transfer is the major contribution to the energy transfer in **PZDN**, triggering the following **NBD** → **QC** conversion.”

3. Showing the energy profile of photochemical reaction of **PZDN-PZDQC** in manuscript would be very helpful for readers.

Reply: We gratefully appreciate the reviewer for his/her valuable suggestion. Accordingly, the diagram of energy profile has been added in the revised manuscript as **Fig. 1b**, which is also shown below for the reviewer’s convenience. Note that it is plausible that the **PZDN** → **PZDQC** photoconversion is a stepwise process for each **NBD** → **QC** conversion, which has been reported in an **NBD** → **QC** system with two **NBD** units attached. (*Adv. Sci.* **2019**, *6*, 1900367). Unfortunately, during the photolysis we could not find the ¹H NMR signals attributed to intermediate where only one **QC** is formed, plausibly due to its short lifespan.

Figure 1b. The proposed energy profile for the photochemical reaction of **PZDN**→**PZDQC** process.

To address the above viewpoint regarding the newly added **Figure 1b**, the following statement has been added in the revised text (see page 5, line 10).

“As depicted in Fig. 1b, the **PZDN** → **PZDQC** photo-conversion is a stepwise process for each **NBD** → **QC** conversion, the mechanism of which has been reported in an **NBD** → **QC** system with two **NBD** units attached. (*Adv. Sci.* **2019**, *6*, 1900367). Unfortunately, during the photolysis we could not find the ¹H NMR signals

attributed to intermediate where only one **QC** is formed (see Fig. 5), plausibly due to its short lifespan.”

Reviewer 2: In this manuscript, Chou et al. report synthesis of three norbornadiene (NBD) derivatives by exploiting thermally activated delayed fluorescence molecules (TADF). The TADF may act as a photosensitizer, storage unit and signal transducer to optimize solar thermal energy storage in the novel molecules. The NBD and its derivatives have been widely studied for MOST. Although the molecular design is interesting, these results cannot advance the progress of solar thermal energy storage materials. I do not suggest acceptance of the manuscript at the present format.

Reply:

We are grateful to the reviewer for his/her valuable opinion. However, we cannot but bring different viewpoint regarding the comment of “**NBD** and its derivatives being used for molecular solar thermal (MOST) systems have been widely studied ----“. Up to this stage, **NBD** and its derivatives are still one of the most popular and reliable systems that are used for studying the MOST effect. This is mainly due to its simplicity in structure, stability in chemical storage, facile chemical derivations and convenient integration into molecular composites suitable for various studies. Support of these advantages can be found in the recent review article entitled “Solar Energy Storage by Molecular Norbornadiene–Quadricyclane Photoswitches: Polymer Film Devices“(Adv. Sci. **2019**, 1900367).

In this study, we proposed a new concept in that solar energy can be harvested nearly equal to the triplet state energy of **NBD** by the use of TADF molecules, maximizing the exploitation of solar energy in the case of **NBD** → **QC** conversion. The success mainly lies in the unique property of TADF molecules **PZDN** and **PZTN**, in which the energy difference between S_1 and T_1 states, i.e., ΔE_{S-T} , is nearly zero. Accordingly, any fine-tuning of the S_1 energy state observed from the UV-vis absorption spectroscopy can directly compare with the triplet state of **NBD** to determine if energy transfer is favorable. This makes possible mutual tuning the **NBD** derivatives and TADF molecules toward longer wavelength. This seminal study of **PZDN** and **PZTN** thus not only proves the concept, it also extends the current hot-topic TADF to the solar thermal energy chemical storage incorporating triplet-state as the photo-conversion mechanism.

1) In Figure 3b,c, I cannot find the emission in the degassed state (maybe purple curve?).

Reply: We are sorry for the awkward drawing. This is mainly due to the overlap of

two curves in color. We have thus replaced the “line” with “line + symbol” for aerated and degassed condition. The new Fig. 3 (now Fig. 2 in the revised version) is also depicted below for the reviewer’s convenience:

Fig. 2. Absorption and emission spectra of PZDN(a), PZTN(b) and PZQN(c) in cyclohexane ($\lambda_{\text{ex}} = 405 \text{ nm}$). Note that the arrow represents the increment of emission intensity along with photolysis time.

2) *The urgent topic of MOST materials should be performed in condensed state, not in*

the solution state.

Reply: We thank the reviewer for pointing out this issue. We agree with the reviewer in that it is imperative for this community to design a device with high energy-storage density, e.g., a large molecular weight-portion of energy storage unit, similar to those of the bridging or substituent (push-pull) units (*Adv. Sci.* **2019**, *6*, 1900367; *Acc. Chem. Res.* **2020**, *53*, 1478–1487; *J. Am. Chem. Soc.* **2020**, *142*, 12256–12264), or devices with high concentration of energy-storage molecule, instead of diluting them into polymer host, which reduces the energy density (*Adv. Energy Mater.* **2018**, *8*, 1703401; *Adv. Sci.* **2021**, 2103060).

Therefore, as suggested by the reviewer, we have conducted independent photo-conversion experiments on **PZDN** in pure solid state (solid film) or imbedded in polymer films including polyethylene(**PE**), polystyrene(**PS**), polyvinyl chloride(**PVC**) or poly(methyl methacrylate)(**PMMA**). Unfortunately, all tested films suffer from severe intermolecular aggregation even in low doping concentration of even less than 0.1 wt%, as indicated by the mismatches of excitation spectra (monitored at different emission wavelength) examined for **PZDN@PS** film (~0.1 wt%) shown in **Supplementary Fig. 31** and below.

Supplementary Figure 31. Excitation spectra of PZDN@PS (0.1 wt%). Note that “em 450” refers to the monitored emission wavelength at 450 nm.

As shown in **Supplementary Fig. 31**, we observe a red-shifted excitation spectrum upon monitoring at 600 nm emission, compared to that when monitoring the emission at 450 nm. The emission wavelength dependent excitation spectra render evidence of intermolecular interactions and thus possible aggregation. This aggregation effect is more significant in high concentration polymer films and pure **PZDN** powder. The molecular aggregation results in lowering the triplet state energy, which is

unfortunately lower than that of the **NBD**'s triplet state, prohibiting the Dexter-type triplet-triplet energy transfer efficiency. As a result, we are unable to observe the changes in population lifetime (monitored at emission peak) upon irradiating 405 nm-light in pure solid **PZDN** film, **PZDN** doped **PE** film or **PMMA** film, where the solubility of **PZDN** is low and aggregation becomes dominant. As for doping **PZDN** in **PVC** and **PS** films, the presence of aggregation seems to be suppressed and there is still a small portion of photo-conversion for **PZDN** monomer, as implied by the increase of population lifetime during the photolysis. The corresponding kinetic traces and emission spectra as a function of irradiation time are shown in **Supplementary Fig. 32** and **33**, which are also depicted below for the reviewer's convenience.

Supplementary Figure 32. (a) Kinetic trace and (b) emission spectra of **PZDN@PVC** (0.1 wt%) as a function of 405 nm-irradiation time.

Supplementary Figure 33. (a) Kinetic trace and (b) emission spectra of **PZDN@PS** (0.1 wt%) as a function of 405 nm-irradiation time.

Clearly, in both **PVC** and **PS** hosts, the emission spectrum gradually blue shifts due to the **PZDN** \rightarrow **PZDQC** conversion of monomer. Additionally, **PZDN@PVC** has a better conversion efficiency than in **PS**, indicating more homogeneous dispersion of

PZDN in **PVC** film. Furthermore, we managed to access the reversibility between **PZDN** and **PZDQC** in both **PVC** and **PS** through irradiating the 405 nm light (3 W LED lamp, 15 minutes to ensure full conversion of **PZDN**→**PZDQC**, marked as red dot and red line in **Supplementary Fig. 34, 35**) and heating at 95°C in a clean oven for an hour (back-conversion of **PZDQC**→**PZDN**, marked as blue dot and blue line in **Supplementary Fig. 34, 35**). The results of photo-thermal conversion cycles are shown in **Supplementary Fig. 34** and **35** for **PZDN@PVC** and **PZDN@PS**, respectively.

Supplementary Figure 34. The durability test for **PZDN@PVC** (●) ⇌ **PZDQC@PVC** (●) (0.1 wt%). (a) Emission intensity monitored at 520 nm. The corresponding (b) emission profiles and (c) population decays during the photo-thermal conversion cycles 0, 1 and 2.

Supplementary Figure 35. The durability test for **PZDN@PS** (●) ⇌ **PZDQC@PS** (●) (0.1 wt%). (a) Emission intensity monitored at 500 nm. The corresponding (b) emission profiles and (c) population decays during the photo-thermal conversion cycles 0, 1 and 2.

Plot (a) of **Supplementary Fig. 34** and **35** shows the intensity monitored at emission

peak, while (b) and (c) refer to the corresponding emission spectra and decay profile in the photo-thermal conversion cycles 0, 1 and 2. The results show that **PZDN** in **PVC** solid films, albeit good photo-conversion efficiency at the first cycle, suffers from severe aggregation and polymerization (*Adv. Energy Mater.* **2018**, *8*, 1703401) during the next light irradiation and heating, resulting in a dramatic decrease in emission intensity and shortening in the decay kinetic trace. As for **PZDN@PS**, the trend in emission intensity holds for 3 cycles (increases after irradiation, decreases after heating), which again verifies **PS** to be a more suitable host for **NBD/QC** system (*Adv. Sci.* **2019**, *6*, 1900367). Note that the anomalous increase in intensity (2nd red dot to 3rd blue dot) is caused by the inhomogeneous thickness of the film, where the steady state measurements for each cycle deviates from the focused excitation source.

Regardless of the device inferior performance in solid, which is a common phenomenon for **NBD** related works (*Adv. Energy Mater.* **2018**, *8*, 1703401), however, this work emphasizes a new concept of combining TADF molecules with **NBD** moieties, which paves a new way for probing the efficiency of **NBD** relevant energy storage. Further practical performance in solid device may be boosted through structural modification, such as introducing steric hindrance to avoid severe aggregation and elongation of **NBD** absorption toward red with matched TADF molecules, so that the sunlight harvesting can be maximized.

We have added a relevant statement before the conclusion section, which is also attached below for the reviewer's convenience.

“Last but not the least, we realized that the urgent topic of **MOST** materials should be performed in condensed state. Therefore, we have conducted independent photo-conversion experiments on **PZDN** in pure solid state (solid film) and **PZDN** imbedded in various polymer films including polyethylene (**PE**), polystyrene (**PS**), polyvinyl chloride (**PVC**) and poly(methyl methacrylate) (**PMMA**). Unfortunately, all tested films suffer severe intermolecular aggregation even in low doping concentration of 0.1 wt% and even less, and hence inferior conversion efficiency and durability. Relevant works are in the Supplementary Fig. 31-35 with detailed elaboration.”

In the conclusion section we also added a statement, written below.

“Further practical performance in solid device may be boosted through structural modification, such as introducing steric hindrance substituents to avoid severe aggregation and elongation of **NBD** triplet state energy toward red, along with spectrally matched TADF molecules, so that the sunlight harvesting can be maximized.”

3) *The small S1-T1 energy gap offers advantage in optimizing solar excitation*

wavelength, however it only covers a narrow arrange in the solar spectrum.

Reply: In this study, we have demonstrated that solar energy can be harvested nearly equal to the triplet state energy of **NBD**, which is ~470 nm, by the use of TADF molecules. This approach has maximized the solar energy in the case of **NBD** → **QC** conversion. The success mainly lies in the unique property of TADF molecules **PZDN** and **PZTN**, in which the energy difference between S_1 and T_1 state, i.e., ΔE_{S-T} , is nearly zero. Accordingly, any fine-tuning of the S_1 energy state observed from the UV-vis spectroscopy can be directly compared with the triplet state of **NBD** to determine if energy transfer is favorable. Furthermore, the proceeding of the reaction can be precisely monitored by the kinetics of TADF, which is unique in the photo-chemical storage system. The result makes possible mutual tuning of the **NBD** derivatives and TADF molecules toward longer wavelength, so that broader solar energy can be harvested based on this strategy. This seminal study of **PZDN** and **PZTN** thus not only proves the concept but also extends the current hot-topic TADF to the solar thermal energy chemical storage that incorporates triplet-state as the photo-conversion mechanism.

4) Only DSC was provided in Supplementary Figure 24, which is not enough for solar thermal energy storage materials. The MOST device should be studied based on charging and recharging processes.

Reply: We are grateful to the reviewer for his/her valuable comment and have demonstrated the durability of degassed **PZDN** ⇌ **PZDQC** cycles, shown in **Fig. 7** and below. The experiment was done by monitoring the emission intensity at 495 nm after 405 nm irradiation to complete the **PZDN**→**PZDQC** process, where the emission intensity is marked as red dot, followed by complete thermal reverse (**PZDQC**→**PZDN**) process at 95°C, marked as the blue dot. As a result, **PZDN** shows great fatigue resistance after 5 charging and recharging cycles.

Fig. 7. The durability of degassed PZDN (●) ⇌ PZDQC (●). The experiment proceeded in cyclohexane and monitored by the emission intensity at ~495 nm.

Reviewer 3: *The Manuscript "A New Approach Exploiting Thermally Activated Delayed Fluorescence Molecules to Optimize Solar Thermal Energy Storage" by Pi-Tai Chou and co-workers considers a molecular photoswitch system capable of converting photon energy into stored chemical energy – so-called molecular solar thermal systems. The novelty of the molecular system is related to the chromophore design, that incorporates a TADF component into the molecular system. Further, the energy storing capability of the system is augmented by the attachment of several NBD/QC units to the central chromophore unit. The basic idea of the manuscript is to drive the photoisomerisation through the triplet manifold via energy transfer from a TADF unit. The introduction is well written with extensive references to the literature. The description of the synthesis is well written, and I appreciated the pedagogic description of the challenges in the synthesis design together with actual synthesis results. The experimental work is complemented with DFT based modelling, trying to understand the positioning of the orbitals. The system is further studied using a combination of time resolved spectroscopic techniques. All together, I think that this is an exciting manuscript that deserves publication in nat. commun. Pending some revisions as noted:*

Major Comments

1. The fluorescence is competing with the photo-conversion, what is the quantum yield of the two processes?

Reply: We thank the reviewer for pointing out this valuable issue. Instant ($t \sim 0$) fluorescence quantum yields (Q.Y.) for **PZDN**, **PZTN** and **PZQN** in cyclohexane were measured in comparison to Coumarin 480 in methanol (Q.Y. = 0.87), and the results are added in Table 1. Furthermore, we carried out the dynamic simulation work (see **Supplementary Fig. 18, 19**) of three titled compounds, the results of which are listed in **Supplementary Table 1**. According to the simulated rate constants, the triplet-triplet energy transfer efficiency (**PXZ-TRZ** core \rightarrow **NBD** moiety) can be deduced by equation (1).

$$\text{Energy transfer efficiency: } \frac{k_{ET}}{k_{ET} + k_{RISC} + k_P} \quad (1)$$

where k_P is the decay rate from T_1 to S_0 , k_{RISC} is the reverse intersystem-crossing rate from T_1 to S_1 , and k_{ET} is the energy transfer rate of from T_1 of **PXZ** or **TRZ** to **NBD**'s T_1 . As a result, the energy transfer efficiency of **PZDN**, **PZTN**, and **PZQN** are calculated to be 59.4%, 14.3%, and 3.7%, respectively. On the one hand, the results can be tentatively rationalized by the distance between **PXZ-TRZ** core and the energy acceptor **NBD** moiety, which is shorter in **PZDN**, resulting in more efficient Dexter type energy transfer. On the other hand, the energy transfer efficiency for **PZQN** is rather small of 3.7%. This is simply due to the thermodynamically unfavorable energy transfer in **PZQN** (*elaborated in the text*).

As suggested by the reviewer, the following paragraph has been added into the revised manuscript:

“From numerical aspects, we then carried out the dynamic simulation based on several valid assumptions^{45,46} (see Supplementary Fig. 18, 19 and corresponding elaboration). These assumptions include a millisecond-scale-decay of the lowest excited triplet state (T_1). Also, the prompt decay rate of the lowest excited singlet state (S_1) is approximated by the population decay of S_1 . The deduced kinetic rates are then tabulated in Supplementary Table 1. In an attempt to quantify the photo-conversion efficiency (Q.Y._{eff}), we further define this term with equation (1) expressed as

$$\text{Q.Y.}_{eff} = \frac{k_{ET}}{k_{ET} + k_{RISC} + k_P} \quad (1)$$

where k_{ET} , k_{RISC} and k_P refer to the rate constants of triplet-triplet energy transfer, reverse inter-system crossing (T_1 to S_1) and the radiative plus non-radiative decay of T_1 , respectively. As a result, the simulated photo-conversion efficiency is 59.4% for **PZDN** and 14.3% for **PZTN** and 3.7% in **PZQN**. The results on the one hand can be rationalized by the distance between **PXZ-TRZ** core and the energy acceptor **NBD**

moiety, which is shorter in **PZDN**, resulting in more efficient Dexter type energy transfer than that in **PZTN**. On the other hand, the energy transfer for **PZQN** is thermally unfavorable (*vide supra*) and hence the associated rate is drastically reduced. This trend correlates well with the change in fluorescence quantum yield before and after light illumination (see Table. 1).”

Minor comments:

1) The spelling and language should be checked, e.g. in the abstract: “triple state of” should be “triplet state”

2) “fMOST” should read “MOST”

3) “implies as closer energy”

Reply: Thanks for the reviewer’s careful inspection on the text. We have explicitly corrected the errors pointed out by the reviewer and those we found in the revised manuscript.

4) To help the presentation, I suggest to make a table that summarizes the properties of the 3 molecular systems, e.g. Amax, A onset, QY(photoconversion), QY(emission), T0.5, Estorage(J/mol), Estorage (J/kg). such a table would make comparison with other systems much more straight forward, see e.g. review by G Han: <https://doi.org/10.1039/D1TC01472B>

Reply: We are grateful to the reviewer for his/her valuable suggestion. As suggested by the reviewer, we prepare a table (Table 1) that includes the absorption peak, absorption onset, fluorescence peak, phosphorescence onset, ΔE_{ST} , Q.Y.(photo-conversion), QY(emission), molecular weight, $E_{storage}(J/mol)$, $E_{storage}(J/kg)$ and $\tau_{1/2}$. Table 1 is also attached below for the reviewer’s convenience.

“Table 1. MOST properties in solution phase

Name	Abs. peak ^a (nm)	Abs. onset ^a (nm)	Fluo. peak ^a (nm)	Phos. onset ^b (nm)	ΔE_{ST} ^c (kcal/mol)	Q.Y. ^d (%)	Q.Y. ^e (%)	MW (g mol ⁻¹)	$\Delta H_{storage}$ (kJ mol ⁻¹)	$E_{storage}$ (J g ⁻¹)	$\tau_{1/2}$ ^f (days)
PZDN	405 nm	465 nm	495 nm	454 nm	0.85	59.4	11.0 ^g 20.5 ^h 66.9 ⁱ	578.23	162	280	76
PZTN	405 nm	465 nm	495 nm	454 nm	0.76	14.3	14.7 ^g 37.8 ^h 71.4 ⁱ 20.0 ^g	934.43	7.97	8.53	- ^j
PZQN	430 nm	495 nm	525 nm	481 nm	1.97	- ^k	66.1 ^h 66.1 ⁱ	988.40	- ^k	- ^k	- ^k

^a Absorption (Abs.) and fluorescence (Fluo.) spectra measured at 298 K in cyclohexane. ^b Phosphorescence (Phos.) spectra measured at 77 K in cyclohexane. ^c ΔE_{ST} is calculated by the energy difference between fluorescence and phosphorescence peak under 77 K. ^d Quantum yield for photo-conversion is calculated by plugging in the simulated rate constants (see Supplementary Table 1) into equation (1). ^e Fluorescence quantum yield measured at 298 K in cyclohexane. ^f Half-life of QC is obtained through extrapolating the Arrhenius plot (Supplementary Fig. 25) to 298 K. ^g Measured in aerated condition before irradiation. ^h Measured in degassed condition before irradiation. ⁱ Measured in degassed condition after irradiation. ^j **PZTQC** \rightarrow **PZTN** is too slow to have practical measurements at 95 °C. ^k Energy transfer can be ignored.”

5) *The figures could in general be more refined, the NMR spectra are formatted in different ways (fig. 6 vs fig. 7, fig. 5 looks terrible with the legend text, etc. please check the journal guidelines, this is a quality journal, there should be quality figures.*

Reply: We are thankful to the reviewer for his/her valuable comments. Accordingly, the two figures of NMR spectra have been updated with an identical format (see **Fig. 5** and **Fig. 6** in the revised manuscript). Also, **Fig. 4** has been refined in both color scale and legend text. Rainbow color has been changed into “Land and Sea” color hue. For the reviewer’s convenience, please see **Fig. 4, 5** and **6** attached below.

Figure 4

Figure 5

Figure 6

6) Why only one compound in fig 8? I suggest to put this in the SI and measure for all compounds. The suggested table can give the needed summary. Check meaningful digits!!

Reply: As suggested by the reviewer, **Fig. 8** has been moved to the SI as **Supplementary Fig. 24**. The reverse thermal conversion for **PZTQC** is too slow to perform practical measurements; thus adding catalyst, CoTPP, is required to complete the process (see **Supplementary Fig. 28**). As for **PZQQC**, there is negligible photoisomerization (**PZQN** → **PZQQC**) taking place under the same condition of 405 nm irradiation. Hence, we did not access the rates of reverse thermal conversion for both **PZTQC** and **PZQQC**. A relevant summary was elaborated in the revised text (see page 6) and below for the reviewer's convenience.

“These results show that **PZDQC** possesses a significant **QC** → **NBD** reversed reaction at relatively low temperature 50-70 °C, which is the main reaction channel before generating any side products. Therefore, the result shows that **PZDN** ↔ **PZDQC** interconversion is highly reversible. In comparison, **PZTQC** → **PZTN** conversion is much slower so the certain branching reaction is competitive, yielding side products (*vide infra*) with the relatively low reversibility.”

Also, numbers in Figures have been revised into significant digits to have meaningful digits.

7) DSC measurements of the energy storage should ideally be provided for all 3 compounds, and the measurements repeated at least 2 times for each, due to the uncertainty of DSC measurements.

Reply: As suggested by the reviewer, DSC measurements of both **PZDQC** and **PZTQC** were carried out twice to reduce the uncertainty. These results were added in the **Supplementary Fig. 29, 30** and below for the reviewer's convenience.

Supplementary Figure 29. DSC thermograms shows the heat release peaks for the thermal back-conversion of PZDQC to the corresponding PZDN. The measurement was repeated twice.

Supplementary Figure 30. DSC thermograms shows the heat release peaks for the thermal back-conversion of PZTQC to the corresponding PZTN. The measurement was repeated twice.

As for **PZQQC**, we did not carry out the DSC experiment because **PZQN** is merely used to illustrate the tuning of its triplet state energy lower than that of NBD, which results in a negligible photo-conversion efficiency.

REVIEWERS' COMMENTS

Reviewer #1 (Remarks to the Author):

With this version, the authors appropriately address the reviewer's comments and revised their manuscript accordingly. Now I am confident that the manuscript is ready for the acceptance for publication in Nature Communications as is.

Reviewer #2 (Remarks to the Author):

I suggest acceptance of the revised manuscript at the present format since all my concerned issues have been addressed.

Reviewer #3 (Remarks to the Author):

The authors have done an excellent job in improving the manuscript, according to the comments of the reviewers,

In my view, the reviewer 2 is overly negative about the manuscript.

The combination of TADF chromophore and the NBD is clearly new, and it brings new functionality to the system by redshifting absorption, without sacrificing storage time (which the authors perhaps are unaware of or could have highlighted more).

I recommend publication in the current form.

RESPONSE TO REVIEWERS

Reviewer 1: With this version, the authors appropriately address the reviewer's comments and revised their manuscript accordingly. Now I am confident that the manuscript is ready for the acceptance for publication in Nature Communications as is.

Reply: We are very grateful to the reviewer for his/her positive comments on this manuscript.

Reviewer 2: I suggest acceptance of the revised manuscript at the present format since all my concerned issues have been addressed.

Reply: We are delighted that our revision has reached the reviewer's criterion.

Reviewer 3: The authors have done an excellent job in improving the manuscript, according to the comments of the reviewers. In my view, the reviewer 2 is overly negative about the manuscript. The combination of TADF chromophore and the NBD is clearly new, and it brings new functionality to the system by redshifting absorption, without sacrificing storage time (which the authors perhaps are unaware of or could have highlighted more).

Reply: We are thankful for the reviewer for his/her appreciation on this manuscript. This work emphasizes on a novel concept that incorporates TADF molecules into energy storage system that maximizes the harvest of solar spectrum, and simultaneously remains a long energy storage time. In all, we appreciate every comments that help refining this work into a finer form.